# USP15 in Cancer and Other Diseases: From Diverse Functionsto Therapeutic Targets

**DOI:** 10.3390/biomedicines10020474

**Published:** 2022-02-17

**Authors:** Yan-Chi Li, Song-Wang Cai, Yu-Bin Shu, Mei-Wan Chen, Zhi Shi

**Affiliations:** 1Department of Cell Biology & Institute of Biomedicine, MOE Key Laboratory of Tumor Molecular Biology, Guangdong Provincial Key Laboratory of Bioengineering Medicine, National Engineering Research Center of Genetic Medicine, College of Life Science and Technology, Jinan University, Guangzhou 510632, China; liyanchi99@stu2019.jnu.edu.cn (Y.-C.L.); sub7@jnu.edu.cn (Y.-B.S.); 2Department of Thoracic Surgery, The First Affiliated Hospital of Jinan University, Guangzhou 510632, China; songwangcai@yahoo.com; 3State Key Laboratory of Quality Research in Chinese Medicine, Institute of Chinese Medical Sciences, University of Macau, Macau 519000, China; mwchen@um.edu.mo

**Keywords:** USP15, deubiquitinatase, ubiquitin, target, cancer

## Abstract

The process of protein ubiquitination and deubiquitination plays an important role in maintaining protein stability and regulating signal pathways, and protein homeostasis perturbations may induce a variety of diseases. The deubiquitination process removes ubiquitin molecules from the protein, which requires the participation of deubiquitinating enzymes (DUBs). Ubiquitin-specific protease 15 (USP15) is a DUB that participates in many biological cell processes and regulates tumorigenesis. A dislocation catalytic triplet was observed in the USP15 structure, a conformation not observed in other USPs, except USP7, which makes USP15 appear to be unique. USP15 has been reported to be involved in the regulation of various cancers and diseases, and the reported substrate functions of USP15 are conflicting, suggesting that USP15 may act as both an oncogene and a tumor suppressor in different contexts. The importance and complexity of USP15 in the pathological processes remains unclear. Therefore, we reviewed the diverse biological functions of USP15 in cancers and other diseases, suggesting the potential of USP15 as an attractive therapeutic target.

## 1. Introduction

Protein degradation plays an important role in the physiological activities of cells. Disorders in protein degradation can cause tumors and some neurodegenerative diseases [1]. The main pathways of protein degradation in eukaryotic cells are ubiquitin-proteasome and autophagy. Autophagy is the process of transporting components to be degraded, which includes not only proteins, but also dysfunctional or redundant organelles, as well as lysosomes for catabolism. Ubiquitin can also participate in the signaling of the lysosomal degradation pathway [2]. Ubiquitination degrades proteins through the ubiquitin–proteasome system and through selective autophagy [3]. The proteasome is a catalytically active and ubiquitous protease, andits most common function is to degrade proteins through the 26s ubiquitin–proteasome system [4]. The 26s proteasome is composed of 20s core particles and 19s regulatory particles. Recent studies have found that the 20s proteasome degrades oxidatively damaged proteins, and unfolds independently of adenosine triphosphate (ATP)/ubiquitin [5,6].

Ubiquitin is a highly conserved peptide molecule, composed of 76 amino acids that post-translationally mark proteins for degradation. Ubiquitin-dependent proteolysis involves the sequential activation of three enzymes, as well as the attachment to target proteins. In the event of ATP supply, the ubiquitin-activating enzyme E1 delivers the ubiquitin molecule to the ubiquitin-conjugating enzyme E2; subsequently, the ubiquitin-ligase enzyme E3 attaches the ubiquitin molecules conjugated at E2 to the target protein, and, finally, this ubiquitinated protein is specifically degraded by the proteasome [7,8,9,10]. Occasionally, multiubiquitylation requires the additional activity of certain ubiquitin-chain elongation factors, such as the E4 enzyme. For example, yeast UFD2 binds to oligo-ubiquitylated substrates (proteins modified by only a few ubiquitin molecules) and catalyzes the multiubiquitin chain assembly in collaboration with E1, E2, and E3 [11,12]. The E4 enzyme has also been shown to be required for highly sustained prolongation [13]. Single or multiple ubiquitin molecules can be conjugated to the substrate protein via their lysine residues (K6, K11, K27, K29, K33, K48, and K63) or N-terminal methionine residues (M1) [14,15], where K48-linked and K63-linked chains are the most well-studied for directing substrate proteins for proteasomal and lysosomal degradation, respectively [16]. K48-linked ubiquitination is usually associated with proteasome-mediated degradation, while K63-linked ubiquitination is involved in autophagy and targets pure proteasomes [17,18,19]. Ubiquitination often affects protein stability, localization, activity, and interactions, and is widely involved in important physiological processes, including cell apoptosis [20,21,22], the cell cycle [23,24], DNA damage repair [25,26], ribosome biogenesis [27], stress responses [28], the induction of inflammatory responses [29], and transcriptional regulation [30]. The ubiquitination process is reversible. The removal of ubiquitin molecules from the ubiquitinated protein, and stabilizing the substrate protein, is known as the deubiquitination process [31,32], which requires the involvement of deubiquitinating enzymes (DUBs). Most DUBs have a clear regulatory role in human cancers and diseases [33]. For example, the loss of ubiquitinC-terminalhydrolaseL1 (UCHL1) activates mitochondrial autophagy in Parkinson’s disease [34], and the increased expression of cylindromatosis (CYLD) inhibits tumor inflammation and angiogenesis to prevent the development and progression of skin squamous cell tumors [35]. Ubiquitination and deubiquitination have been found to be dysregulated in different types of cancer, possibly due to mutations, deletions, or amplifications [36,37,38].

## 2. DUBs

DUBs are proteases which remove ubiquitin molecules from the ubiquitinated proteins. DUBs can regulate ubiquitination in four distinct mechanisms: (1) processing polyubiquitin molecules into single ubiquitin molecules, (2) recycling ubiquitin molecules, (3) preventing E3 ligase-mediated ubiquitin conjugation, and (4) cleaving polyubiquitin chains [39,40,41]. According to the structure of the catalytic domain, DUBs are divided into seven classes: ubiquitin-specific proteases (USPs), Josephin and JAB1/MPN^+^(MJP), JAB1/MPN/Mov34 metalloenzyme (JAMM), ubiquitin C-terminal hydrolases (UCHs), ovarian tumor proteases (OTUs), and the two recently discovered MIU-containing novel DUB (MINDY) and zinc finger-containing ubiquitin peptidase 1 (ZUP1) [41,42,43,44,45]. Except for ZUP1, which is a metalloprotease, the rest are cysteine proteases [46,47]. USPs are the most numerous family of DUBs [48]. USP15 is a member of the USPs family and is involved in various cellular processes. Here, we will provide a comprehensive overview of the role of USP15 in various cancers and other diseases, which will help us better understand the biological functions of USP15 and provide strategies for USP15 as a therapeutic target.

## 3. Structure and Characteristics of Ubiquitin-Specific Protease 15

The human USP15 gene is located on chromosome 12q14.1 and shares 60.5% sequence identity and 76% sequence similarity with the human homolog (UNP/Unph/USP4) of the mouse Unp proto-oncogene [49]. USP15 and USP4 have an overlap in their function and deubiquitination substrates. Fertility testing in mice showed that USP15 and USP4 have a sufficient functional redundancy to rescue inactivated mutations in a reciprocal manner [50]. The evolutionarily conserved skipping of exon 7 during the transcription of USP15 mRNA results in the deletion of a short element in the interregional link of USP15, thus producing two major isoforms, 1 and 2 [51], consisting of 981and 952 amino acid residues, respectively [52]. Kotani et al.found differences in the substrate preference between the two isoforms of USP15, that is, RNF213, a ubiquitin ligase associated with Moyamoya disease, is primarily recognized and deubiquitinated by isoform 1 [51], indicating the functional differences between the isoforms. In addition, the micronucleus phenotype can be rescued by USP15 isoforms when USP15 is depleted in U2OS cells [51]. Moreover, USP15 isoform 1 is preferentially upregulated in a set of non-small cell lung cancer cell lines, suggesting that heterodimer imbalance may contribute to genomic instability in cancer [53].

The domain structure of USP15 is composed of a catalytic region and a non-catalytic region. The catalytic region contains the typical catalytic triad, Cys-269, His-862, and Asp-879 [47], which participates in the formation of part of the active site of USP15. The mutation of Cys-269 to Ser suppresses the enzymatic activity of USP15 [54]. However, the catalytic triad of USP15 is in an inactive conformation, with the catalytic cysteine being far away from the catalytic histidine, which is atypical. There is a protease domain that harbors an approximately 300-amino acid insertion containing an ubiquitin-like (UBL) domain. The N-terminal’s non-catalytic domain includes the ubiquitin-specific protease domain (DUSP) that specifically binds to the substrate protein and the other UBL domain [55,56]. The USP15 UBL domain interacts with the DUSP domain through a UBL surface area that is rarely used as recognition site [57] (Figure 1a). A zinc (Zn) finger is discovered in the detailed analysis of USP15 and the related USPs sequences, which indicates that USP15, and the related USPs, require a functional Zn-finger to adopt a conformation that is needed for the binding and cleavage of is peptide-bond-linked polyubiquitin chains [58,59]. The structure of USP15, predicted by the Alpha Fold, is shown in Figure 1b.

## 4. Function of USP15 in Cancers and Other Diseases

In humans, USP15 is highly expressed in endocrine tissues, the gastrointestinal tract, the liver, the gallbladder, bone marrow, lymphoid tissues, etc. (Figure 2).USP15 is upregulated in a variety of cancers, including glioblastoma [60], breast cancer [60], ovarian cancer [61], multiple myeloma [62], prostate cancer [63], gastric cancer [64,65], pancreatic ductal adenocarcinoma [66,67], and chronic myeloid leukemia [68]. However, the USP15 gene is deleted in 25.37% of pancreatic cancers and 10.9% of glioblastomas, implying a potential tumor suppressive function of USP15 in these cancers [69].USP15 is involved in various cellular processes, such as cell proliferation, cell invasion, apoptosis, autophagy, the cell cycle, genome integrity, transcription regulation, the immune response, and others. The complex and diverse biological functions indicate that USP15 is a critical protein that needs to be studied more comprehensively. The following sections will review, in detail, the important functions of USP15 in cancers and other diseases.

### 4.1. Proliferation, Migration, and Invasion

USP15 has been reported to promote or inhibit tumorigenesis and its progression in different cancers. Previous reports have proved that USP15 is highly expressed in breast cancer. Interleukin-1 receptor type 2 (IL1R2) binds with, and enhances, USP15 to deubiquitinate BMI1at the K81 residue, thereby promoting the breast cancer cell lines SUM159 and MB231, as well as their proliferation and invasion [70]. Moreover, Xia et al. revealed that USP15 promotes the proliferation of the breast cancer cell lines T47D and MCF-7 by deubiquitinating and stabilizing the estrogen receptor α (ERα) in vitro and in vivo, which is related to the cell cycle regulation, rather than cell apoptosis [71]. ERα can bind to RNA to regulate the growth of tumor cells and the efficacy of the anti-cancer drug, tamoxifen [72]. These findings indicate that USP15 may be a potential therapeutic target for breast cancer. Furthermore, Zhou et al. found that the expression of USP15is upregulated in the multiple myeloma cell lines, and USP15 overexpression enhances cell proliferation and inhibits cell apoptosis in multiple myeloma cell lines, such as RPMI 8226 and U266. Mechanistically, USP15 deubiquitinates and stabilizes NF-κB inhibitor α (IκBα), and USP15 is, in turn, positively regulated by NF-κB through increasing the activity of the USP15 promoter [62]. Most patients with hematological malignancies and multiple myeloma have developed resistance to immunomodulatory drugs. Cereblon (CRBN) is an E3 ubiquitin ligase and a direct protein target for the immunomodulatory and antiproliferative activities of lenalidomide and pomalidomide, and USP15 antagonizes the ubiquitination of CRBN target proteins, such as glutamine synthetase, thereby preventing their degradation [73]. There are many reports that USPs participate in the development of gastric cancer and can be used as independent prognostic markers for gastric cancer patients. The silence of USP15 inhibits cell proliferation and invasion in the gastric cancer cell lines BGC-823 and MKN-45 in vivo and in vitro, and USP15 overexpression shows the opposite result. In terms of mechanisms, USP15 overexpression promotes the nuclear expression of β-catenin in gastric cancer cells, thereby activating the Wnt/β-caten in signaling pathway to promote the malignant progression of gastric cancer, suggesting that USP15 may be an oncogene of gastric cancer [65]. However, there has been another report stating that USP15 deubiquitinates IκBα to inhibit the NF-κB pathway, thereby blocking the proliferation and invasion of the gastric cancer cells lines SGC7901 and MKN45 [74].These data indicate that the functions of USP15 are complicated, and functions in the same types of cancer may be different, so continuous exploration is needed. Silencing USP15 also inhibits cell proliferation and invasion in glioblastoma cell lines U87-MG and U251-MG, indicating that USP15 may be a potentially effective treatment target for glioblastoma [75]. The knockdown of USP15 results in the decrease in the E3 ubiquitin ligase MDM2 protein in the melanoma cell line A375 and the colorectal cancer cell line HCT116, indicating that USP15 plays an important role in stabilizing MDM2. USP15 stabilizes MDM2 by the deubiquitination and regulation of p53 function, which, in turn, negatively regulates T-cell activation by targeting the degradation of the transcription factor, the nuclearfactorofactivatedT-cells2 (NFATc2), ultimately regulating both tumor growth and anti-tumor immunity [76]. Interestingly, the transforming growth factor β (TGFβ) enhances the production of USP15 by promoting the upregulation of USP15 translation, rather than transcription, let alone inhibiting USP15 degradation. USP15 interacts with, and stabilizes, p53 through deubiquitination, and it regulates the transcriptional activity of p53, thereby promoting the expression of p21 to inhibit human osteosarcoma cell line U2OS proliferation [77]. On the contrary, Padmanabhan et al. found that USP15 controls the protein expression of p53-R175H, but not p53 WT, through an ubiquitin-mediated lysosomal pathway in ovarian cancer cells [63]. Recent reports indicate that USP15 is highly expressed in liver cancer tissues compared with normal tissues, and the knockdown of USP15 inhibits cell proliferation and enhances cell apoptosis in the hepatocellular carcinoma cell lines SNU449 and Hep3B [78]. Furthermore, USP15 deubiquitinates and stabilizes kelch like ECH associated protein 1 (KEAP1) to degradate nuclear factor erythroid 2-related factor 2 (NRF2), which is enhanced by xanthine oxidoreductase, thereby leading to the accumulation of reactive oxygen species and changes in the oxidative environment, ultimately inducing cell apoptosis and suppressing the propagation of hepatocellular carcinoma [79]. Additionally, Chen et al. found that the expression of USP15 is upregulated in the rat model of epilepsy, and the suppression of USP15 activates the NRF2/HO-1 pathway, inhibiting the glutamate-induced oxidative damage in the mouse hippocampal neuron cell line HT22 [80]. USP15 also deubiquitinates and stabilizes NRF1 to promote NRF1-induced proteasome gene expression, thereby maintaining proteostasis [81]. USP15 can deubiquitinate the insulin receptor substrate 2 (IRS2), and IRS2 monoubiquitination increases insulin-like growth factor 1(IGF1) signaling in HEK293 cells [82]. However, IRS2 only binds to USP15 after it is connected to the ubiquitin molecule, thereby blocking IGF1-dependent IRS2 phosphorylation to weaken IGF-I signaling in prostate cancer PC-3 cells [83,84]. USP15 interacts with DEK and the kinesin family member 15 (KIF15) and stabilizes DEK to facilitate the proliferation of the leiomyosarcoma cell lines SK-UT-1 and SK-LMS-1 [85]. USP15 also deubiquitinates and stabilizes SMAD-specific E3 ubiquitin protein ligase 2 (SMURF2) at the K734 residue to affect the catalytic activity of SMURF2, which targets the TGFβ receptor complex for ubiquitin-mediated degradation, thereby regulating TGFβ-dependent cell migration [86]. Furthermore, USP15 deubiquitinates and stabilizesthe type I TGFβ receptor (TGFβRI) to enhance the TGFβ signaling pathway and promote glioblastoma multiforme cell proliferation and invasion [60,87]. In addition, USP15 deubiquitinates and stabilizes TGFβRI to enhance TGFβ/SMAD signaling and to promote the proliferation and invasion of hypertrophic scar-derived fibroblasts, suggesting that USP15 could be a potential target for the treatment of hypertrophic scars [88]. USP15 deubiquitinates and interacts with the extracellular-signal-regulated kinase 2 (ERK2), but it does not regulate the stability of ERK2, resulting in an increase in the pERK1/2 levels to further enhance the TGFβ/SMAD2 signaling pathway and suppress osteoarthritis progression. Moreover, overexpression of USP15 can repair the joint plane in the rat osteoarthritis model, showing that USP15 plays an important role in preventing cartilage damage in vivo and in vitro [89]. Additionally, USP15 deubiquitinates and interacts with the eukaryotic translation initiation factor 4A1 (EIF4A1) to promote the proliferation and migration of keratinocytes and to accelerate the re-epithelialization during wound healing in mice [90].

### 4.2. Apoptosis

The occurrence of tumors is not only related to abnormal cell proliferation and differentiation, but also to abnormal apoptosis. USP15 has been reported to participate in various ways to promote cell apoptosis. USP15 deubiquitinates and stabilizes caspase-6 to increase cell apoptosis in the chronic myeloid leukemia cell K562, and miR-202-5p inhibits the expression of USP15 by directly targeting the 3’ untranslated region (3’UTR) of USP15 to suppress cell apoptosis [91]. USP15 also deubiquitinates and stabilizes procaspase-3 to promote paclitaxel-induced apoptosis in the human cervical cancer cell line HeLa [92]. Furthermore, Yu et al. analyzed the mRNA expression levels of nine DUB family members, including USP15, in eight normal tissues and 10 degenerative nucleus pulposus tissues, and found that only USP15 was highly expressed in the nucleus pulposus tissues, compared to the normal tissues. Subsequent findings have shown that USP15 deubiquitinates and stabilizes FK506 binding protein 5 (FKBP5), thereby promoting the apoptosis of nucleus pulposus cells by inhibiting the phosphatidylinositol-3-kinase (PI3K)/Akt pathway [93].Wang et al. also found that USP15 may dynamically target the endolysosomes on mitochondria and impair the mitochondrial outer membrane permeabilization functionality in the human breast cancer cell line MCF7, thereby participating in the process of apoptosis signaling [94].The role of USP15 in the regulation of apoptosis provides insights for the treatment of cancer and other diseases.

### 4.3. Autophagy

USP15 not only participates in the ubiquitin–proteasome system, but also plays a role in autophagy. Parkinson’s disease is a neurodegenerative disease which mainly occurs in the elderly, and its pathogenesis is mainly due to mitochondrial dysfunction. The E3 ligase PARKIN has been reported toplay a critical role in the autophagy clearance of defective mitochondria, which mediates the clearance of a variety of mitochondrial proteins after being activated [34,95]. USP15 can deubiquitinate PARKIN to impair mitochondrial autophagy, suggesting that USP15 inhibition could be a therapeutic strategy for Parkinson’s disease, caused by reduced PARKIN levels [96,97]. It has been reported that the expression of USP15 increased in the miR-26a knockdown mouse model, and miR-26adirectly targets USP15 to regulate cardiomyocyte autophagy, as well as playing a protective role in myocardial ischemic injury [98]. The above studies suggest that USP15 may be a potential target for the treatment of some diseases by participating in the regulation of autophagy.

### 4.4. Cell Cycle

The COP9 signalosome (CSN) affects the stability of microtubule filaments during the cell cycle [99]. USP15 has been reported to bind to CSN and protect CSN-related components [58]. Therefore, USP15 may be involved in cell cycle regulation. USP15 has been reported to accumulate on the newly synthesized RE1-silencing transcription factor (REST) in the early G1, playing an important role in the correct separation of the mitotic chromosome. Andrew et al. verified that USP15 deubiquitinates and stabilizes REST, but it does not antagonize the β-TrCP-mediated degradation of REST, which quickly replenishes REST when mitosis exits at the beginning of a new cell cycle [100,101,102]. Xiaet al. analyzed the cell cycle and the apoptosis-related proteins by flow cytometry after silencing USP15 in breast cancer cells, demonstrating that USP15 promotes breast cancer cell proliferation through the cell cycle [71]. Therefore, USP15 may regulate cancer progression by participating in the cell cycle.

### 4.5. Genome Integrity

It has been found that DNA is damaged in the USP15 knockout mice, and USP15 is phosphorylated by the ataxia telangiectasia mutated (ATM) kinase at the S678 residue, suggesting that USP15 plays a role in homologous recombination [54]. USP15 is enrolled by the mediator of DNA damage checkpoint 1 (MDC1) to the DNA double-strand break to deubiquitinate BRCA1 associated with RING domain 1 (BARD1), enhancing the BARD1-HP1γ interaction and the BRCA1/BARD1 retention at the DNA double-strand break, as well as reducing the PARP inhibitor sensitivity in breast and ovarian cancer cells [54]. There is a decrease in the survival rate and weight loss in the USP15 knockout mice, and USP15 interacts with and stabilizes the DNA repair factor FUS to promote the self-renewal of hematopoietic stem cells [68]. The squamous cell carcinoma antigen, recognized by T-cell 3 (SART3), shuttles USP15 to the nucleus through the nuclear localization sequence and promotes H2B deubiquitination in free histones by USP15, thereby affecting transcription, replication, and the DNA repair processes [103]. At the same time, SART3 also helps USP15 recruit free histones [104]. It has been reported that the knockdown of USP15 results in the expression decrease in the topoisomerase II α (TOP2A) protein, instead of mRNA, in the human non-small cell lung cancer cell line A549, indicating that USP15 regulates TOP2A to maintain genome integrity [53]. Moreover, USP15 cannot protect genome integrity and inhibit micronucleus formation by interfering with mitosis after being phosphorylated at the S229 residue [53]. These studies suggest that USP15 maintains genomic integrity by regulating DNA damage repair, replication, and transcription.

### 4.6. Transcriptional Regulation

USP15 acts in certain diseases through the transcriptional regulation of target genes. USP15 deubiquitinates and redistributes terminal uridylyl transferase 1 (TUT1) from the nucleolus to the nucleoplasm, resulting in the stabilization of U6 snRNA, thereby regulating the neurodegenerative phenotype [105]. USP15 also deubiquitinates and opposes R-SMAD monoubiquitylation to prevent R-SMADs recognizing the promoters involved in the differentiation of mesenchymal stem cells from osteoblasts [106]. In addition, USP15 has been reported to regulate dynamic protein-to-protein interactions in the spliceosome through the deubiquitination of the pre-mRNA-processingfactor31 (PRP31) and the DUSP-UBL domains of USP15 and PRP31, which only interact in the present of SART3 [107]. USP15 also deubiquitinatesand stabilizes the transcription factor, sulfurimitation1 (SLIM1), to regulate the hypertrophic responses of cardiac muscle in mice [108].

### 4.7. Immune Response

USP15 participates in the regulation of multiple signal pathways and plays a key role in cellular immunity and the inflammatory response. USP15 deubiquitinatestetmethylcytosinedioxygenase2 (TET2) at the K1299 residue to inhibit the TET2 activity and to increase the expression of chemokines induced by interferon γ (IFNγ), resulting in a longer lifespan of mice [109]. There is a reduced *Listeria* load in the liver, a more enlarged spleen, and a higher level of serum IFNβ in the USP15 knockout mice [76]. Moreover, the TRAF-interacting protein with a forkhead-associated domain B (TIFAB) binds to the catalytic domain of USP15 to enhance USP15-deubiquitinating MDM2 at the K48 residue, thereby decreasing the expression of p53 [110]. USP15deubiquitinates and stabilizes the E6 protein of HPV to inhibit the E6-mediated p53 degradation in HPV-positive HeLa cells [111]. In addition, the E6 protein can block the deubiquitination of the tripartitemotifcontaining25 (TRIM25) by USP15, which ultimately leads to the degradation of TRIM25, thereby inhibiting the retinoic acid-inducible gene 1 (RIG1)-mediated antiviral signal transduction in the cervical cancer cell line C33A [112,113]. USP15 negatively regulates the activation of naive CD4+ T-cells and the generation of IFNγ, as well as the redundancy of IFNγ in USP15-deficient mice, which increases the expression of programmed death ligand 1 (PD-L1) and the C-X-C motif chemokine ligand 12 (CXCL12) to promote the immunosuppressive tumor microenvironment processed by immunosuppressive tumors induced by methylcholantrene [114]. USP15 deubiquitinates and stabilizes TANK-binding kinase 1(TBK1) at the K63 residue to inhibit type I IFN production in vitro and in vivo, which is mediated by the ubiquitin-conjugating enzyme UBE2S, thereby increasing virus replication [115]. However, USP15 also deubiquitinates and interacts with TRIM25 and prevents the linear ubiquitin chain assembly complex (LUBAC) and the E3 ligase-dependent degradation of TRIM25. IFNβ promoter activation caused by Sendai virus (SeV) infection is decreased after the knockdown of USP15 in HEK293T cells. The deubiquitination of TRIM25 by USP15 enhances the production of type I IFN and inhibits the replication of RNA viruses, resulting in an antiviral response [116,117]. Therefore, USP15 is not only recruited by UBE2S to suppress type I IFN production by deubiquitinating TBK1, but it also stabilizes TRIM25 to promote TRIM25 and RIG1-dependent type I IFN production, which is due to USP15 targeting different proteins and participating in different signaling pathways. Although USP15 does not interact with the hepatitis C virus (HCV) proteins and does not participate in innate immune responses, knockout of USP15 inhibits the translation of HCV RNA and the accumulation of lipid droplets in the hepatocellular carcinoma cell line Huh7, as well as the addition of palmitic acids, which partially recovers the production of infectious viral particles, suggesting that USP15 regulates HCV propagation by affecting viral RNA translation and lipid metabolism [118]. The above function of USP15 in the immune response suggests that USP15 may be a potential therapeutic target for cancer immunotherapy.

### 4.8. Others

USP15 interacts with TGFβ-activated kinase 1 (TAK1)-binding protein (TAB) under the stimulation of the tumor necrosis factor α(TNFα), thereby inhibiting the degradation of TAB2/3 through the deubiquitination pathway, as well as inhibiting lysosomal degradation, or inhibiting the selective autophagy pathway. Therefore, USP15 can positively regulate the TNFα- and interleukin-1β-induced NF-κB activation [119,120]. USP15 deubiquitinates and stabilizes the hepatitis B virus X (HBx) protein in the human liver cancer cells Huh7, HepG2, and Hep3B [121]. Additionally, USP15 regulates the degradation of the human immunodeficiency virus (HIV) Nef and Gag proteins, thereby blocking the replication of HIV, indicating that USP15 can eliminate viral proteins by mediating protein degradation [122]. USP15 also deubiquitinates and stabilizes the bone morphogenetic protein (BMP) type I receptor ALK3 to promote BMP signal transduction [123]. Additionally, USP15 deubiquitinates and stabilizes E3 ubiquitin ligase BRAP, and BRAP, in turn, enhances the ubiquitination of USP15 to regulate MAPK signal transduction [124]. It has been reported that the endoplasmic reticulum-located E3 ubiquitin ligase Hrd1 can also ubiquitinate and interact with USP15 at the K21 residue in HEK293T cells, which does not cause USP15 to be degraded, but it loses its activity for IκBα deubiquitination during bacterial infection, leading to the excessive activation of NF-κB. Moreover, the knockout of Hrd1 in macrophages protects mice against septic shock induced by lipopolysaccharide, and the silence of USO15 in Hrd1-deficiency macrophages recovers the decreased production of interleukin-6 [125]. USP15 is also involved in the vesicle transport process. The interaction of the E3 ubiquitin ligase ring finger protein 26 (RNF26) and USP15 was detected in HEK293T cells. The overexpression of USP15 can deubiquitinate the substrate, sequestosome1 (SQSTM1) of RNF26; that is, USP15 and RNF26 are functionally competitive. Furthermore, RNF26 inhibits vesicle transport through ubiquitination, while USP15 releases trapped vesicles through deubiquitination, thereby achieving the cyclic regeneration of vesicle transport [126]. These studies provide sufficient information to enrich the biological functions of USP15 and to develop USP15-targeted therapeutic systems. The function, expression and target of USP15 in various cancers and other diseases were summarized in Table 1 and Figure 3.

## 5. Conclusions

This review combines recent USP15-related research to illustrate the multi-faceted role of USP15 in cancer and other diseases. It is clear from the above discussion that USP15 is upregulated and plays an important role in the pathogenesis of cancer and other diseases. The involvement of USP15 in intracellular processes, such as apoptosis, autophagy, and the immune response has led to USP15 being an attractive therapeutic target. The comprehensive proteomic strategies accelerate the identification of USP15 substrate proteins. Evidence suggests that USP15 stabilizes oncoproteins by deubiquitination, thereby promoting proliferation in most cancers, including breast cancer, multiple myeloma, gastric cancer, glioblastoma, etc. However, some studies also point to USP15 acting as a negative regulator in certain tumor growth, which complicates the role of USP15 in tumors, suggesting that USP15 is a critical and promising target for tumor therapy. Therefore, the more comprehensive study of USP15 is needed to refine its biological function. Additionally, there is an urgent need to develop USP15-specific inhibitors to investigate its role in disease. Recently, a preclinical small-molecule inhibitor of USP15, USP15-Inh, selectively damages leukemia progenitor cells through re-engaging homeostatic redox responses [127]. Mitoxantrone has also been reported to inhibit the activity of USP15 to a certain extent [55]. The development of USP15 inhibitors undoubtedly provides a bright future for USP15-targeted therapy.

## Figures and Tables

**Figure 1 biomedicines-10-00474-f001:**
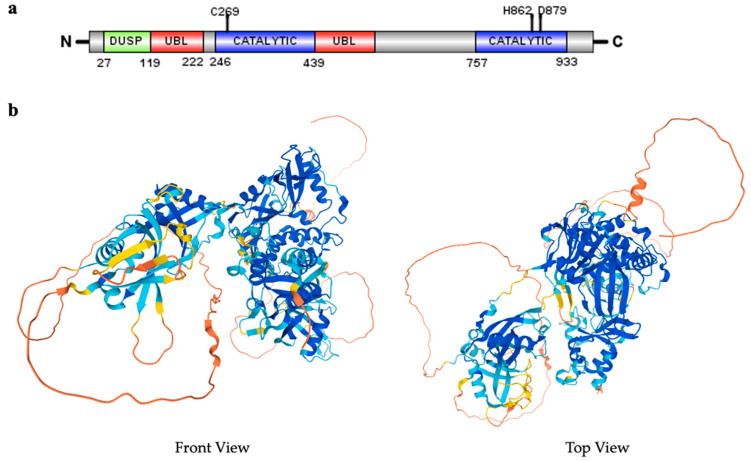
Schematic diagram of USP15 structure. (**a**) Schematic diagram of the domain composition of USP15 (isoform 2); (**b**) USP15 structure in front and top view directions. Available online: https://alphafold.ebi.ac.uk/entry/Q9Y4E8 (accessed on 26 October 2021).

**Figure 2 biomedicines-10-00474-f002:**
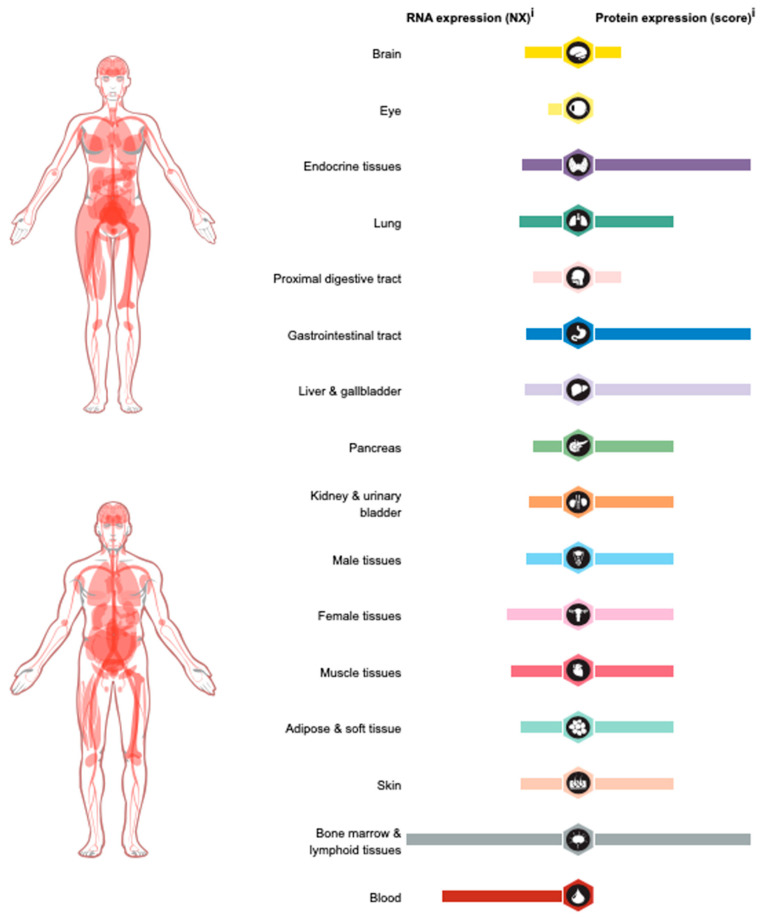
The expression of USP15 RNA and proteins in various human tissues. Available online: https://www.proteinatlas.org/ENSG00000135655-USP15/tissue (accessed on 8 November 2021).

**Figure 3 biomedicines-10-00474-f003:**
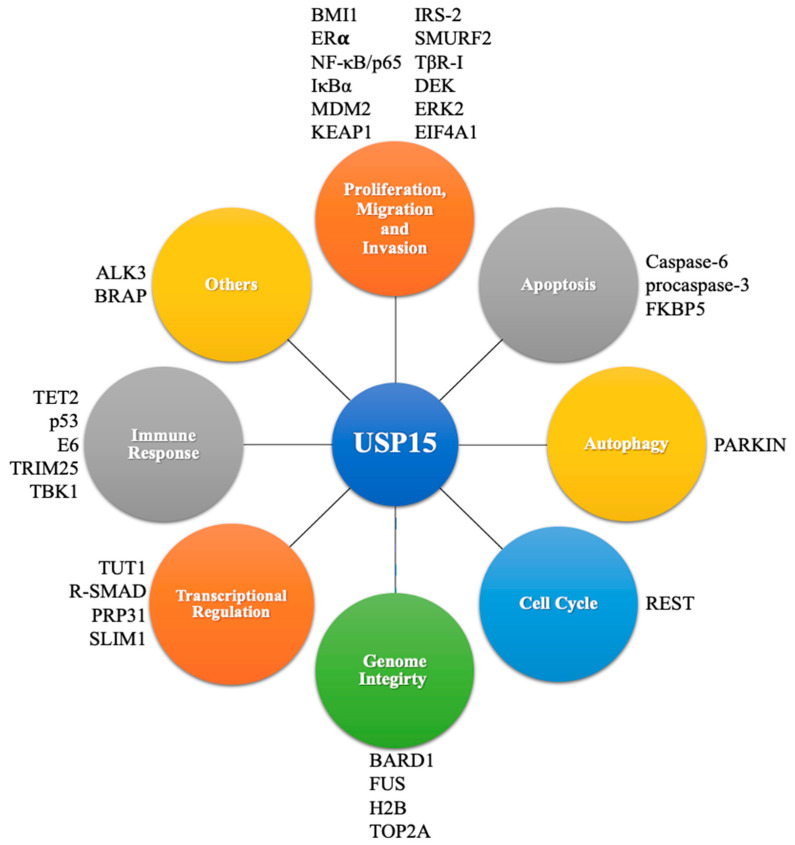
Cellular processes and substrates involved in USP15.

**Table 1 biomedicines-10-00474-t001:** The function, expression, and target of USP15 in various cancers and other diseases.

Function	Cancer/Disease	Expression of USP15	Target
Proliferation, migration, and invasion	Breast cancer	Up [60]	BMI1 [70], ER𝛂 [71]
	Multiple myeloma	Up [62]	NF-κB/p65 [62]
	Gastric cancer	Up [65]	IκBα [74]
	Colon cancer	Up [76]	MDM2 [76], p53 [77]
	Hepatocellular carcinoma	Up [78]	KEAP1 [79]
	Prostate cancer	Up [63]	IRS-2 [82,83]
	Glioblastoma multiforme	Up [60]	SMURF2 [86], TβR-I [60,87]
	Leiomyosarcoma		DEK [85]
	Hypertrophic scar		TβR-I [88]
	Osteoarthritis		ERK2 [89]
	Wound healing		EIF4A1 [90]
Apoptosis	Chronic myeloid leukemia	Up [68]	Caspase-6 [91]
			Procaspase-3 [92]
			FKBP5 [93]
Autophagy	Parkinson’s disease	Up [96]	PARKIN [96,97]
Cell cycle			REST [100,101,102]
Genome integrity	Ovarian cancer	Up [61]	BARD1 [54]
	Leukemia	Up [68]	FUS [68]
			H2B [104]
			TOP2A [53]
Transcriptional regulation	Neurodegenerative diseases		TUT1 [105]
			R-SMAD [106]
			PRP31 [107]
			SLIM1 [108]
Immune response	Melanoma	Up [76]	TET2 [109]
	Human papillomavirus		E6 [111], TRIM25 [116,117]
			TBK1 [115]
Others			ALK3 [123]
			BRAP [124]

## Data Availability

The data herein presented are available in this article.

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
