# Peer review of "USP15 in Cancer and Other Diseases: From Diverse Functionsto Therapeutic Targets"

_biomedicines, 2022, doi:10.3390/biomedicines10020474_

Round 1

Reviewer 1 Report

I do not see much improvement of the manuscript quality/flow in this revised version. It is highly unlikely that this work has been reviewed by an English native speaker.

Author Response

Thanks for the referee’s careful reading. We have already checked the whole manuscript and corrected the mistakes of English language and style in the text.

Reviewer 2 Report

     In their review entitled “USP15 in cancer and other diseases: from diverse functions to therapeutic targets” Li and colleagues depicted the diverse biological functions of USP15 in cancer and other diseases.  The authors attempt to describe the cancer-related functions of USP15 to better understand its complexity in promoting or suppressing cancer proliferation.

This review is interesting and focuses its observations on USP15 in relation to cancer. On the other hand, the review lacks clarity, some sentences are too complicated and difficult to understand. The review should be improved in terms of written English language and clarity of ideas to make the manuscript more accessible.

In addition, I have several comments to improve the quality of the manuscript:

  • The authors did not discuss the presence or the role of USP15 isoforms nor the cellular distribution of USP15 in paragraph 3. In addition, the authors did not comment on the observation that USP15 is deleted in some cancers (for example: USP15 gene is deleted in 25.37% of pancreas cancer patients…). The authors should discuss these observations.
  • I suggest to remove from the section 3; the paragraph from line 128-147 and put it either in section 4.1 or in the section 4.8 others since the authors started to describe the function of USP15 in cancer proliferation and HIV replication… while this paragraph is dedicated to the structure and characteristics of USP15.
  • I missed a more critical look from the authors and their point of view on this functional diversity of the USP15. Is there a redundancy of the isoforms? Is there a compensation with the other DUBs?

In several sections, the authors end with a non-meaningful phrase, such as section 4.5 “USP15 does not directly target gene integrity, it can participate in tumor progression by indirectly regulating gene integrity….”

Can the authors elaborate more their conclusions?

Author Response

Comments:1. The review lacks clarity, some sentences are too complicated and difficult to understand. The review should be improved in terms of written English language and clarity of ideas to make the manuscript more accessible.

Response 1:We agree with the referee. We have already checked the whole manuscript and amended the incorrect tense and voice in the text.

Comments:2. The authors did not discuss the presence or the role of USP15 isoforms nor the cellular distribution of USP15 in paragraph 3. In addition, the authors did not comment on the observation that USP15 is deleted in some cancers (for example: USP15 gene is deleted in 25.37% of pancreas cancer patients…). The authors should discuss these observations.

Response 2:We appreciate the referee’s for this great suggestion.We have already added these information into the content in the revised manuscript.

Comments:3. I suggest to remove from the section 3; the paragraph from line 128-147 and put it either in section 4.1 or in the section 4.8 others since the authors started to describe the function of USP15 in cancer proliferation and HIV replication… while this paragraph is dedicated to the structure and characteristics of USP15.

Response 3:Thanks for the referee’s careful reading.The paragraph from line 128-147 was originally at the beginning of section 4, and we have now moved ittothe section 4.8.

Comments:4.I missed a more critical look from the authors and their point of view on this functional diversity of the USP15. Is there a redundancy of the isoforms? Is there a compensation with the other DUBs?

Response 4:Thanks for the referee’s nice suggestion. We have already added these information, which can be found in the revised manuscript.

Comments:5. In several sections, the authors end with a non-meaningful phrase, such as section 4.5 “USP15 does not directly target gene integrity, it can participate in tumor progression by indirectly regulating gene integrity….”. Can the authors elaborate more their conclusions?

Response 5:Thanks for the referee’s kind suggestion. We have drawn more conclusions in the revised manuscript.